# Mechanical Desensitization of an Al/WO₃ Nanothermite by Means of Carbonaceous Coatings Derived from Carbohydrates

**Pierre Gibot [1],\* , Quentin Miesch [1], Arnaud Bach [1,2], Fabien Schnell [1], Roger Gadiou [2] and Denis Spitzer [1]**

1   Laboratoire des Nanomatériaux pour desSystèmes Sous Sollicitations Extrêmes (NS3E), CNRS-ISL-UNISTRA UMR 3208, Institut franco-allemand de recherches de Saint-Louis (ISL), 5 rue du Général Cassagnou, BP70034, 68301 Saint-Louis, France

2   Institut de Science des Matériaux de Mulhouse (IS2M), CNRS UMR 7361, Université de Haute Alsace (UHA), 15 rue Jean Starcky, BP2488, 68057 Mulhouse, France

\*   Correspondence: pierre.gibot@isl.eu; Tel.: +33-(0)3-89-69-58-77

**Abstract:** Nanothermites show great developmental promise in the near future in civilian, military and aerospace applications due to their tuneable reactive properties (ignition delay time, combustion velocity and pressure release). However, the high mechanical sensitivities of some of these energetic nanocomposites can make transportation and handling of them hazardous. Here, a mechanical desensitization (shock and friction) of an Al/WO₃ nanothermite is successfully obtained by means of carbon adding through the pyrolysis of naturally occurring molecules (carbohydrates). The combustion behaviour of the carbon-based energetic mixtures were also evaluated and a respectable reactivity has been evidenced.

**Keywords:** nanothermites; saccharides; carbon; desensitization; energetic nanocomposites

---

## 1. Introduction

The mixtures of metal and oxide powders at the micro-scale, which are known as thermites, are interesting energetic composites for specific applications, such as in the ordnance domain. The reaction between both components is described as an exothermic oxidoreduction chemical reaction leading to a high adiabatic temperature [1]. The amount of energy released during the reaction is considerable and can reach, in some cases, 20 kJ/cm³. The adiabatic flame temperatures are also very impressive with values ranging from 2000 to 3000 K. However, these traditional micro scale thermites may suffer from low performance, such as poor reactivities, high ignition delay times and low combustion velocities. These intrinsic reactive properties can be widely improved by the exclusive use of nanosized metallic and oxides powders since the chemical reactions are governed by diffusion/transport phenomena at the interphase of the reactants [2]. The shorter the diffusion pathway is, the better the reactive properties. In such a configuration, the thermites, which are referred to as nanothermites or metastable intermolecular composite (MIC) materials, attract much attention [3–10]. These are very promising energetic composites and can advantageously substitute for explosives in many pyrotechnic applications due to their high available energy densities and reaction rates [11–13], while being free of highly toxic metal (mercury and lead) [11–13]. However, the downside of obtaining such reactive properties is that, in most cases, the sensitivities of the thermites to external stresses, such as mechanical ones, are highly increased [14]. In addition, the extreme sensitivity of the nanothermites towards the electrostatic discharge does not need to be demonstrated. Subsequently, such behaviours can be harmful to the expected potential applications of these kinds of energetic composites since

accidents can occur and the safety of the users cannot be assured. The development of desensitized nanothermites that maintain suitable reactivity is; thus, of primary importance. From an international point of view, the United Nations intergovernmental organization has established standards for the transport of explosive goods [15]. The energetic materials are classified with respect to their sensitivity thresholds relating to several stresses, such as impacts, friction, electrostatic discharge, temperature increases, magnetic fields and so on. According to these standards, the energetic compositions of the nanothermites are crucial to their future promise, as expected.

In the literature, a number of investigations have been conducted on mitigating the sensitivities of nanothermites. Aluminium metal was systematically chosen as the fuel and was mixed with different oxide ceramics ($CuO$, $WO_3$, $MoO_3$ and $MnO_2$) [16–24] or a polymer (polytetrafluoroethylene, PTFE) [25–27]. Desensitization can occur due to two different approaches: Either by modifying the aluminium particles (average particle size and passivizing alumina layer) [21,22] or by adding an additive in the traditional binary fuel/oxidizer energetic composites [16–20,23–27]. Regarding this second method, in a large majority of studies, synthetic carbons such as nanotubes, graphene and nanofibres were selected that provided more efficient electrostatic discharge desensitization [16,17,24–27] without effect or mention regarding the friction sensitivity. Now, some nanothermites are very friction sensitive such as $Al/CuO$, $Al/WO_3$ and $Al/MnO_2$ with values of ranged from 8 to 9 N [20,24] that is widely below the 80 N standard required for the transportation of energetic materials on public roads [15]. Maintaining such level of friction sensitivity could restrict practical applications of nanothermites. Since then, the use of cheap amorphous carbon black (soot) was proposed within the $Al/WO_3$ nanothermites to mitigate the friction sensitivity. Efficient mechanical desensitization (impact and friction) was observed with low amounts of carbon additives with a decreased combustion velocity [19]. More recently, two papers have shown the beneficial use of non-carbonaceous additives to mitigate the sensitivities of nanothermites. The first one examined and demonstrated the efficiency of a lubricating molybdenum sulphide material ($MoS_2$) and hexadecane ($C_{16}H_{34}$) as friction desensitizing additives in an $Al/MoO_3$ nanothermite, but no reactive properties were evaluated [23]. The second paper focused on the use of a polyaniline-conducting polymer in an $Al/WO_3$ nanothermite. The polymer proved to be a very promising additive for the friction and electrostatic discharge desensitizations at low amounts (5 wt.%) while maintaining good reactive performance for the investigated energetic formulation [20]. However, the performance of the nanothermite strongly depends on the homogenization/dispersion of the additive within the $Al/WO_3$ energetic composites.

Generally speaking, the adding of additives within nanothermites leads to a mitigated reactive behaviour of the considered energetic systems (decrease of the combustion velocity, longer ignition delay time, etc.) that can thereby limit or divert the thermites of their initial practical applications, but, on other side, it is important to keep in mind that using nanothermites free of any additives is extremely hazardous (handling and transportation). Definitively, the nanothermite desensitization is an emerging research topic where numerous aspects must be explored (additive nature, combustion behaviour and potential applications)

Here, a carbon coating onto the tungsten (VI) oxide nanoparticle surface as a method to mechanically desensitize an $Al/WO_3$ nanothermite is described. The carbon adding was achieved by a fast and simple process that carbonizes sucrose and cellulose molecules. These organic species (saccharides) are naturally occurring molecules that are abundant and low cost, making them promising alternatives in the development of insensitive reactive energetic formulations. The performance of the nanothermites (mechanical sensitivities and reactivity), enriched or not enriched with carbon additives, were characterized.

## 2. Materials and Methods

Sucrose ($C_{12}H_{22}O_{11}$, reagent grade), acetonitrile ($CH_3CN$, 99%) and tungsten (VI) oxide ($WO_3$, specific surface area (SSA) = 27 $m^2$/g, 99.9%) were purchased from Sigma-Aldrich. Sulphuric acid ($H_2SO_4$, 98%) was obtained from Carl Roth. Cellulose microcrystalline (($C_6H_{10}O_5)_n$, reagent grade)



was provided by Alfa Aesar. Aluminium (Al, SSA = 37 m$^2$/g, Al = 57.9 wt.%) nanopowder was purchased from NovaCentrix (Austin, TX, USA). All chemical reagents were used as received.

### 2.1. Tungsten (VI) Oxide/Carbon Composites Synthesis

The weight percent of carbon additives in the WO$_3$/C composite was fixed at approximately 3 wt.% because, for lower values, tungsten (VI) oxide particle surfaces are not completely darkened, especially with the use of a cellulose precursor. For that, the carbonization reactions of sucrose (Equation (1)) and cellulose (Equation (2)) were considered.

$$C_{12}H_{22}O_{11}(s) \rightarrow 12C(s) + 11H_2O(g) \tag{1}$$

$$(C_6H_{10}O_5)_n(s) \rightarrow 6nC(s) + 5nH_2O(g) \tag{2}$$

The WO$_3$/saccharide composites were assessed following two different approaches, depending on the saccharide molecule (sucrose or cellulose) that was used as the carbon source. With the sucrose molecule, the experimental approach can be described as follows. In a round-bottom flask (100 mL), 1 g of WO$_3$ and 0.19 g of sucrose were poured in 50 mL of demineralized water and stirred and ultrasonicated for 30 min. Then, 100 µL of concentrated sulphuric acid (H$_2$SO$_4$) was added and the stirring was maintained for 30 min. The round-bottom flask was placed on a rotating evaporator (100 rpm) and the liquid phase was evaporated at 90 °C under 200 mbar of pressure until a dry powder was obtained. The product was heated up to 250 °C at 5 °C/min and maintained at this temperature for 2 h to complete the drying. Concerning the preparation of the WO$_3$/saccharide composites with the use of cellulose as the carbon source, the experimental method is as follows. 1 g of WO$_3$ and 0.14 g of microcrystalline cellulose were mixed in a mortar. Then, 100 mg of this mixture was inserted in a 5 mL agate grinding jar that was equipped with an agate ball (5 mm, 100 mg) and placed on a mixer-mill apparatus (MM200, Retsch, Haan, Germany). The mixture was then milled for 5 min at a frequency of 25 Hz. The sucrose-based composite was not submitted to this post-treatment since sucrose has high solubility in water and a high physical contact between components is expected after the solvent removal.

The resulting WO$_3$/saccharide composites were heat-treated at 500 °C (5 °C/min, 2 h) under an argon flow (100 mL/min) in order to assess the WO$_3$/carbon composites. This onset temperature was chosen to be low in order to avoid interfacial reactions between the components leading to the synthesis of the under-oxygenated tungsten oxides (WO$_{3-x}$) or tungsten carbide (W$_x$C) phases [28]. The experimental yields in the solid carbon phase were close to 20 and 24 wt.% for the sucrose and cellulose, respectively. These data were lower than the theoretical yields (42.1 and 44.5 wt.% according to Equation (1) and Equation (2), respectively) mainly due to the formation of tars and gaseous carbonaceous species [29,30].

### 2.2. Nanothermite Preparation

The energetic formulations were prepared by mixing the tungsten (VI) oxide and aluminium nanopowders in amounts defined according to the following stoichiometric chemical reaction (3).

$$2Al(s) + WO_3(s) \rightarrow Al_2O_3(s) + W(s) \tag{3}$$

Therefore, an excess of aluminium was preferred in order to enhance the performance of the nanothermites [4]. The excess active aluminium was adjusted by means of the equivalence ratio ($\Phi$) described in Equation (4) and was fixed at a value of 1.4 instead of 1:

$$\Phi = ((F/O)exp.)/((F/O)st.) \tag{4}$$

F/O is the mass ratio of fuel (Al) to oxidizer ($WO_3$) and the subscripts exp. and st. stand for the experimental and stoichiometric ratios, respectively.

The aluminium and the tungsten (VI) oxide powders, which were covered by carbon, were poured into 60 mL of acetonitrile, sonicated and stirred for 1 h in order to break the aggregates and to prepare homogenous mixtures. The solvent was evaporated under 200 mbar of pressure at 80 °C until dry greyish powders were obtained. The powders were then placed in an oven at 80 °C for 4 h. The energetic compositions were labelled as $Al/WO_3/C_x$, where x represents the nature of the biomolecules (*suc* for sucrose and *cell* for cellulose) that were used for the carbon adding within the nanothermite. An $Al/WO_3$ nanothermite without a carbon additive was also prepared as a reference.

## 2.3. Characterization Techniques

The chemical compositions (C and H) of the saccharide-derived carbon were performed by burning at 1050 °C in a helium/oxygen flow with a Thermo Finnigan Flash EA 1112 apparatus (Thermo Fisher Scientific, Waltham, MA, USA). The resulting gaseous products ($CO_2$ and $H_2O$) were separated on a chromatographic column and quantified by using a thermal conductivity detector. The amount of oxygen was calculated with respect to 100% of the material. The structures of the $WO_3$/carbon composites were determined by using an X-ray diffractometer (XRD, D8 Advance, Bruker, Karlsruhe, Germany) operating at 40 kV–40 mA. The diffractograms were collected in the 2 theta range 10–80° with a step size of 0.02° and a time by step of 0.1 s. The weighted amounts of carbon in the $WO_3$/carbon composites were determined by thermogravimetric analysis (SDT Q600 TA Instruments Inc., New Castle, DE, USA). The different composites were calcinated under air at between 25–800 °C with a heating rate of 10 °C/min. A scanning electron microscope (SEM, FEI Nova NanoSEM 450, FEI Company, Hillsboro, OR, USA) working at 10 kV was used to observe the morphology of the $WO_3$/carbon samples that were previously covered by a thin layer of gold (Desk II TSC Denton Vacuum, Moorestown, NI, USA).

The performance of the $Al/WO_3/C_x$ nanothermites was evaluated at once in terms of the sensitivity thresholds (impact and friction) and reactivity (combustion velocity). The impact sensitivity was determined by a Julius Peters BAM Fall hammer apparatus and the threshold value is given in joules (J). The friction sensitivity, whose value is in Newtons (N), was determined by a BAM friction apparatus. More details on the working principles of these apparatuses are mentioned in [20]. The different observed threshold values corresponded to the highest value, for which 6 reaction tests were related to a no-combustion result. The probability of obtaining no combustion at all was 98.4% at that threshold value. All of the experiments were performed according to the standards that were established by NATO [31–33]. Reactivity tests were performed on the nanothermite powders by means of an optical flash ignition, which was inspired by the works of Wang et al. and Ohkura et al. [34,35]. The energy density that was delivered by our laboratory-made equipment, which was equipped with a xenon lamp, was 0.11 $J/cm^2$. The heating temperature and the heating rate were determined to be higher than 400 °C and $5 \times 10^5$ °C/s., respectively, by means of a nanocalorimeter. Typically, 10 mg of powdered energetic nanocomposites were placed on a glass substrate following a predefined contact surface of approximately 50 $mm^2$ in order to expose the same amount of matter to the flash. The glass substrate was placed on the flash igniter device at a respective distance of 1 mm. All the combustion tests were performed in air on all unconfined powdered $Al/WO_3$ nanothermites (with or w/o the carbon additive). The combustion phenomena of the energetic formulations were recorded by using a Photron FASTCAM (Photron, Reutligen, Germany) high-speed camera ($10^3$ frames/s) placed in front of the set-up.

## 3. Results and Discussion

### 3.1. Characterization of the Saccharide-Derived Carbons and the $WO_3$/C Composites

Figure 1B,C shows the X-ray diffraction patterns of the different $WO_3$/carbon composites that were synthesized via pyrolysis at 500 °C of the corresponding $WO_3$/saccharide multi-materials. The

diffraction peaks that are observed for all composites, with respect to the examined angular domain, can be attributed to the $WO_3$ tungsten (VI) oxide phase (Figure 1A). The oxide ceramic crystallizes in the monoclinic structure with a P21/n space group, as reported in the crystallographic card No 01-083-0950. Some of the Miller planes were given on the corresponding figure. No other diffraction peaks were detected, thereby confirming the amorphous character of the carbon component and the fact that the composites are free of any crystalline impurities such as tungsten carbide ($W_xC$) and oxygen-deficient tungsten oxides ($WO_{3-x}$) phases. Absence of any amorphous phases was also confirmed by Fourier transform infrared analysis (Figure S1).

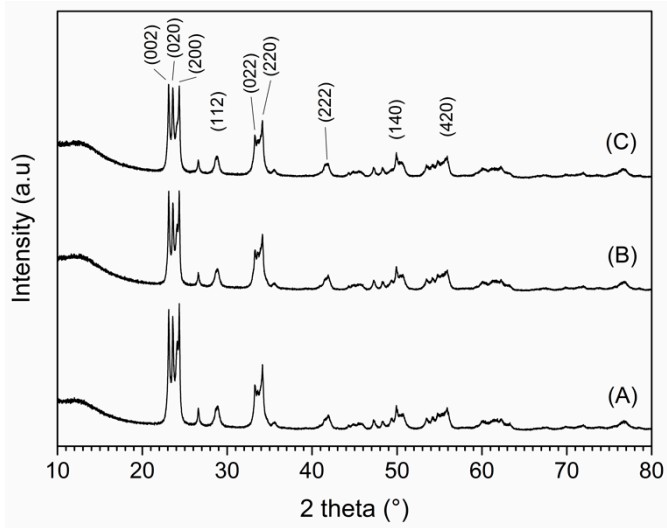

**Figure 1.** XRD patterns of the pure $WO_3$ component (**A**) and the $WO_3/C_{suc}$ (**B**) and $WO_3/C_{cell}$ (**C**) composites synthesized via the pyrolysis of the corresponding $WO_3$/saccharide multi-materials.

The amounts of carbon in the $WO_3$/carbon composites were determined by means of thermogravimetric analyses, as reported in Figure 2. Two weight losses for the $WO_3/C_{suc}$ and $WO_3/C_{cell}$ composites were recorded: below 300 °C and between 300 to 700 °C. While the first weight loss can be attributed to the release of the adsorbed water molecules and embedded solvents on or within the oxide/carbon composites, the second one was interpreted as the oxidation of the carbon-rich phases. The weight losses at the highest temperature ranges are 3.5% and 3.1% for the $WO_3/C_{suc}$ and $WO_3/C_{cell}$ composites, respectively.

To precisely characterize the oxide/carbon multi-materials, the chemical composition of the carbon phases were determined by elemental analysis. Table 1 gathers the data relating to the saccharide molecules after pyrolysis at 500 °C under argon for 2 h. As reported, the resulting products are mainly made of carbon with amounts that ranged from 86 to 82 wt.% with the use of sucrose and cellulose as reactants, respectively. The presence of the hydrogen and oxygen heteroatoms could be explained by the low temperature that was used for the pyrolysis, which caused the incomplete carbonization of carbohydrates. In the literature, temperatures close to 900 °C are classically used [28]. We voluntarily chose a lower pyrolysis temperature to avoid the formation of tungsten derivatives (see "Materials and Methods" Section—Paragraph 2.1), which could affect the combustion behaviour of the energetic formulations that are investigated. To sum up, the $WO_3/C_{suc}$ and $WO_3/C_{cell}$ composites are made of a carbonaceous phase that is globally similar in terms of its elemental composition with an oxygen amount ranging from 11 to 15 wt.%, respectively, which is to the detriment of the carbon amount.

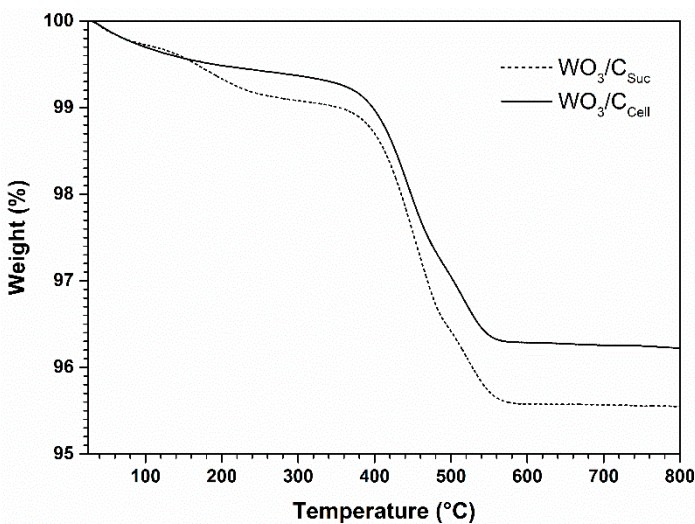

**Figure 2.** Thermogravimetric curves of the $WO_3/C_{suc}$ and $WO_3/C_{cell}$ composites under air between 25 to 800 °C.

**Table 1.** Chemical composition of the saccharide-derived carbon phases. The oxygen amount was calculated with respect to 100% of the material.

|  | **C wt.%** | **H wt.%** | **O wt.%** |
|---|---|---|---|
| $C_{suc}$ | 85.7 | 3.2 | 11.1 |
| $C_{cell}$ | 82.0 | 3.3 | 14.7 |

The SEM images of the tungsten (VI) oxide/carbon composites are presented in Figure 3. Both composites, $WO_3/C_{suc}$ (Figure 3A) and $WO_3/C_{cell}$ (Figure 3B), are made of micron-sized particles with surfaces that appear to be very rough. After careful observation, this roughness corresponds to the highly agglomerated very small particles with very similar sizes to the pristine tungsten (VI) oxide particles that were used for the $WO_3$/carbon composites' preparation (Figure S2). The elemental mappings of the carbon and tungsten chemical elements corresponding to these $WO_3$/carbon composites, which were established by means of energy dispersive X-ray (EDS) spectroscopy, are presented in Figure 3A,B. Obviously, for both composites, the carbon and tungsten are evenly distributed across all microparticles and seem to be intimately mixed. The fact that carbon and tungsten can be observed at identical places suggests the successful coating of the carbon additive onto the tungsten (VI) oxide particles' surface.

This hypothesis was confirmed by means of transmission electron microscopic analysis, as shown in Figure 4, where core-shell structures were observed for both $WO_3$/carbon composites. However, the carbon coating that was achieved from the sucrose precursor seems to be less uniform and homogeneous on the $WO_3$ nanoparticles' surface (Figure 4A) where large carbonaceous matter was sometimes found between the tungsten-based particles, than the carbon-shell issued from the cellulose molecules (Figure 4B).

After mixing with the aluminium nanoparticles in order to prepare the nanothermites according to the experimental section, the amounts of carbon were recalculated as 2.7 and 2.4 wt.% in the $Al/WO_3/C_{suc}$ and $Al/WO_3/C_{cell}$ energetic formulations, respectively (2.3 and 2.0 wt.%, respectively, when taking into account the alumina layer coating the aluminium nanoparticles). The microstructures of both $Al/WO_3/C_x$ energetic formulations were also examined by scanning electron microscopy (SEM) and the corresponding images are shown in Figure 5A,B for the nanothermites that are were enriched with the carbonization products of sucrose and cellulose, respectively. For both $Al/WO_3/C_x$ mixtures, a similar result is obtained, namely, the observation of very fine particles on all examined samples, and when larger particles are seen, they correspond to an agglomeration of smaller ones, as previously

observed for the $WO_3$/carbon composites (Figure 3). The aluminium nanoparticles seem to be well distributed within the two as-prepared nanothermites without leading to microstructural evolutions. The microscopic views of the ternary energetic systems are accompanied by the aluminium, carbon and tungsten elemental mappings (Figure 5—down). In a general way, both composites exhibit effective and successful volume distributions for their three chemical components (except for oxygen). For the tungsten element, some islets describing slightly richer domains in the transition metal can nevertheless be observed. As this trend is observed on the images describing the $Al/WO_3/C_{suc}$ and the $Al/WO_3/C_{cell}$ systems, both as-prepared nanothermites can then be said to exhibit similar homogeneities.

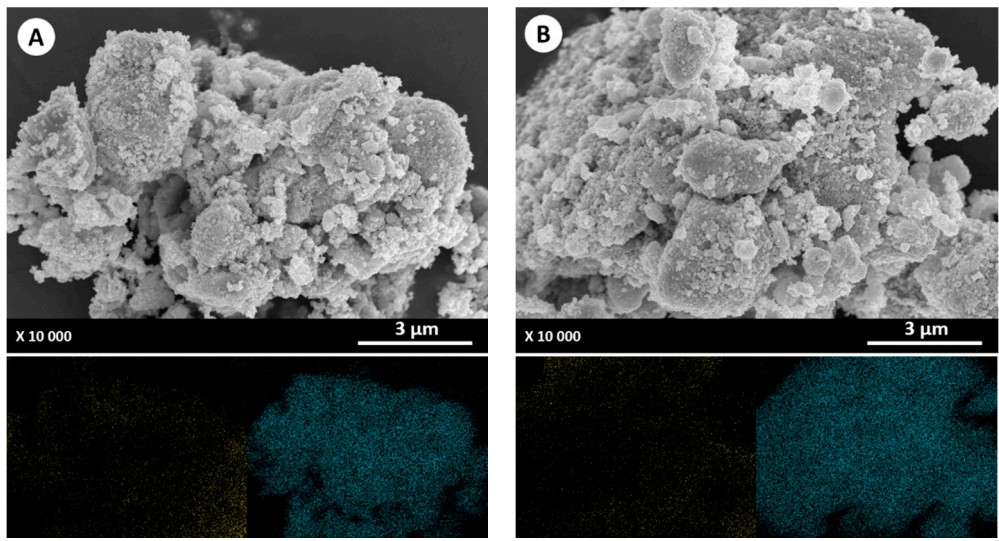

**Figure 3.** SEM views of the (**A**) $WO_3/C_{suc}$ and (**B**) $WO_3/C_{cell}$ composites. Elemental mapping analysis of the carbon (yellow plot) and tungsten (blue plot) are reported for each diffused electron image.

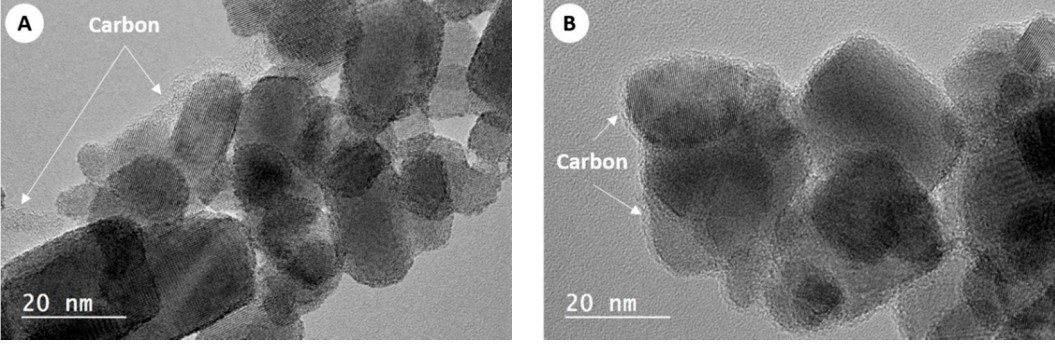

**Figure 4.** TEM images of the (**A**) $WO_3/C_{suc}$ and (**B**) $WO_3/C_{cell}$ composites.

The two $Al/WO_3$ nanothermite mixtures that were prepared with carbon additives were used in their powder forms in order to assess their performances, such as their sensitivity thresholds for mechanical solicitations and their combustive behaviours.

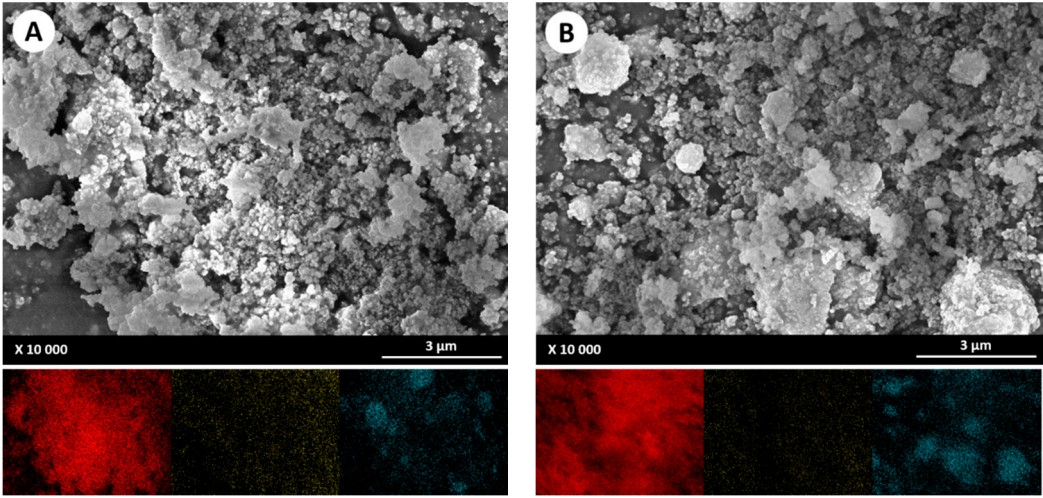

**Figure 5.** SEM views of the (**A**) Al/WO$_3$/C$_{suc}$ and (**B**) Al/WO$_3$/C$_{cell}$ mixtures. The elemental mapping analysis of the aluminium (red plot), carbon (yellow plot) and tungsten (blue plot) are reported for each diffused electron image.

## 3.2. Sensitivities of the Al/WO$_3$/C Nanothermites

The mechanical sensitivities (i.e., impact and friction of the ternary energetic formulations) are presented and compared to the binary Al/WO$_3$ nanothermite in Table 2. The different sensitivity thresholds were determined according to the protocol detailed in [20].

**Table 2.** Mechanical (impact and friction) sensitivity thresholds for the Al/WO$_3$/C nanothermites (weight % of carbon of approximately 2.7 and 2.4 from sucrose and cellulose as the carbon sources, respectively).

|  | Impact (J) | Friction (N) |
|---|---|---|
| Standards | 2 | 80 |
| Al/WO$_3$ | 42.2 | 8 |
| Al/WO$_3$/C$_{suc}$ | >49.6 | 216 |
| Al/WO$_3$/C$_{cell}$ | 47.1 | 324 |

Regarding the impact sensitivity, the Al/WO$_3$ binary formulation can be considered to be insensitive with a threshold value of 42.2 N, which is significantly higher than the standard set at 2 J, as established by the United Nations [15]. The use of carbon as an additive led to a systematic increase of this value with values of >49.6 and 47.1 J for the Al/WO$_3$ nanothermites enriched with carbon coming from sucrose and cellulose carbonization, respectively. Thus, the adding of carbon within an Al/WO$_3$ energetic composite allows one to preserve and to strengthen the insensitivity of the formulation by increasing its impact sensitivity threshold [36].

Regarding the friction sensitivity, the Al/WO$_3$ nanothermite is very sensitive, with a threshold value of 8 N compared with the standard value of 80 N established by NATO [15]. The use of carbon, regardless of its origin (sucrose or cellulose), allowed one to definitely raise the sensitivity threshold value of the Al/WO$_3$ nanothermite, ranging from 216 N (carbon sucrose) to 324 N (carbon cellulose), which ranks the carbon-based ternary nanothermites as moderately sensitive [36].

In a general point of view, regarding the mechanical sensitivities, the very low impact and friction sensitivities of the different Al/WO$_3$/C$_x$ formulations might be explained by the fact that the carbon adding, which leads to the preparation of either a core-shell structure and/or a homogeneous mixture of tungsten (VI) oxide and carbon for the WO$_3$/C$_{cell}$ and WO$_3$/C$_{suc}$ composites, respectively (Figures 3 and 4), allows one to isolate the oxidizer of the reducing metal compound. Consequently, when the ternary energetic formulations are mechanically stressed (impact and friction tests), no hot spots are generated at the aluminium tungsten trioxide interphase because no direct contact exists between Al

and $WO_3$. The ignition of the thermites is then avoided at equivalent stress levels to the ones that are defined for the binary reference $Al/WO_3$ nanothermite. Higher mechanical solicitations are; thus, required to burn the carbon-based energetic compositions.

Compared to the literature dedicated to the desensitization of the nanothermites by means of additives [16,17,19,23–27], our results, which are focused on mechanical sensitivity mitigation, can be compared to only a couple of works [19,20,23,24] since most of them are focused on electrostatic discharge sensitivity mitigation. Moreover, sometimes the nature of the oxidizer component used for the nanothermite formulation is different ($MoO_3$, $MnO_2$, etc.), which may lead to difficult direct comparisons [23,24]. However, due to the necessity to compare our results to the literature, it can be argued that the present results, in a general point of view, perform as well as or better than those in the literature. For instance, with respect to the impact sensitivity threshold, Siegert et al. obtained desensitization ranging from 24.5 to 44.2 J with 29 to 37 wt.% of carbon nanofibres added (depending on the selected method for introducing the additive within the $Al/MnO_2$ nanothermite) [24], and Gibot et al. obtained a threshold value of 20 J with the addition of the polyaniline conductive polymer (5 wt.% within the $Al/WO_3$ energetic system) [20]. Thus, the present results are similar to the results that were obtained by Bach et al. with a 5 wt.% of carbon black as the desensitization agent in the $Al/WO_3$ nanothermites (impact sensitivity > 49.6 J) [19]. Regarding the friction sensitivity threshold values that were obtained in this work (216 and 324 N depending on the carbon source), again, the data are very competitive since they are clearly superior to the results of Kelly et al. (120 N with a 5 wt.% of $MoS_2$ as an additive within an $Al/MoO_3$ nanothermite) [23] and slightly lower and/or on the same order of magnitude as those of [19,20,24], namely, >360 N, with an amount of additive that was almost twice as much as the present work.

### 3.3. Combustion of the Al/WO$_3$/C Nanothermites

The combustion properties of the $Al/WO_3$/carbon nanothermites with added carbon coming from sucrose or cellulose carbonization were compared with the classical $Al/WO_3$ nanothermite behaviour.

The camera flash ignition process has been carried out to investigate the combustion phenomena. The advantages of this method are that it does not depend on the environmental conditions (temperature, pressure, humidity, etc.), it delivers constant thermal energy and it can be performed under air on unconfined powdered energetic systems. Then, all phenomena related to the evolution of the gas through carbon combustion should be avoided. Some selected video images of the combustion of the three investigated energetic nanocomposites are shown in Figure 6.

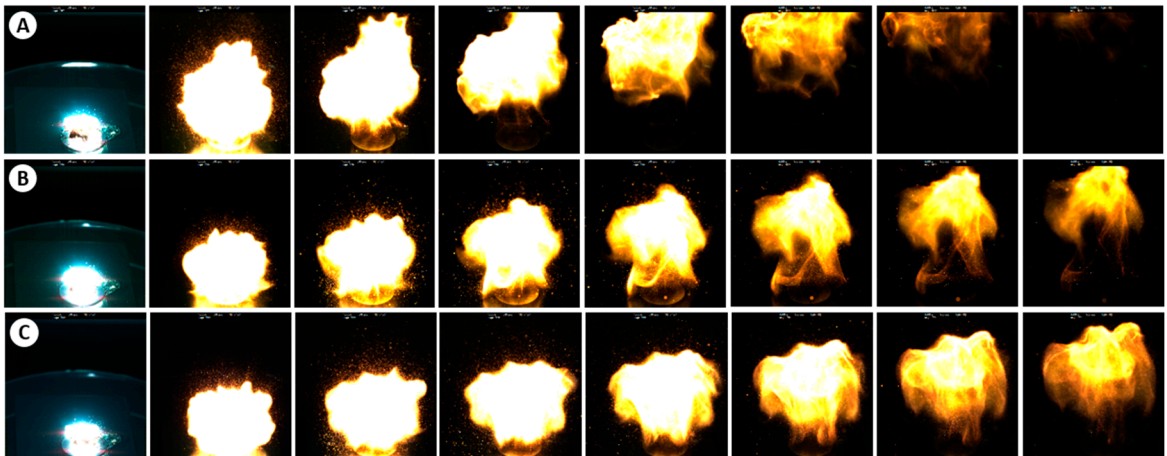

**Figure 6.** Sequences of the combustion images of the (**A**) Al/WO$_3$, (**B**) Al/WO$_3$/C$_{suc}$ and (**C**) Al/WO$_3$/C$_{cell}$ nanothermites. The first image of each series corresponds to the ignition of the energetic powders. The elapsed time between pictures is 200 μs.

In all cases, the energetic compositions completely combusted because no residual solid products (traces) were found on the glass substrate at the end of the tests. An intense white light is observed in the first frames of the recording videos and it takes on a yellow-orange tone at the end of the combustion, expressing a rapid increase in the temperature followed by cooling. This result suggests that the addition of a carbonaceous agent did not lead to a total loss of reactivity for the $Al/WO_3$ nanothermite. Nevertheless, while a clapping sound was heard for the $Al/WO_3$ nanothermite, the combustions of the carbon-doped systems were quieter. Most likely, the latter have a less powerful energetic character than the binary energetic nanocomposite. This mitigation of the combustion energy might be due to the energy consumed by the carbon additive for its own combustion (Equation (5)), leading to a slight decrease of the total released energy according to the examined formulations.

$$2C(s) + O_2(g) \rightarrow 2CO(g) \tag{5}$$

This trend was expected and has been already noted with the adding of other additives within a binary nanothermite [19,20]. Finally, when observing the different sequences of the combustion images of the $Al/WO_3$, $Al/WO_3/C_{suc}$ and $Al/WO_3/C_{cell}$ nanothermites (Figure 6), it might be that the combustion velocity is higher for the binary energetic system (Figure 6A) than for the ternary ones (Figure 6B,C). This hypothesis is based on the fact that for equivalent amounts of energetic matters (w or w/o carbon) that are ignited by means of an optical flash, the yellow-orange colour (synonymous with the cooling step of the energetic formulation) of the incandescent cloud located above the glass substrate is visible at a shorter duration time (~0.8 ms.) of the combustion of the binary $Al/WO_3$ nanothermite, than for the carbon-enriched $Al/WO_3$ nanothermites ($\geq$1.2 ms.). In other words, for the same amount of matter, the total combustion of the undoped mixtures is quicker than that for carbon-doped ones. To the same extent, the combustion velocities of both $Al/WO_3$/carbon ternary nanothermites seem to be relatively similar. However, no more accurate details of the combustion velocity can be provided since it was difficult to evaluate the covered distance by the reaction front as a function of time. Currently, a reflection is conducted to optimize the in-air-combustion tests (without the confinement of nanothermites) and permit an accurate characterization of the combustion velocities of these carbon-doped energetic formulations.

## 4. Conclusions

In summary, an efficient friction desensitization of an $Al/WO_3$ nanothermite while maintaining acceptable reactive properties was achieved using carbon as a desensitizing agent. The carbon additive was obtained by carbonizing the sucrose and cellulose molecules, which are naturally occurring molecules that are abundant and low cost. With a carbon amount slightly lower than 3 wt.% within the nanothermite (with the carbon coming from the sucrose carbonization), the mechanical sensitivity threshold values are about >360 N and >49.6 J for the friction and impact tests, respectively, which are at least 4.5 and 25 times higher, respectively, than the international standard values (80 N and 2 J, respectively). With regard to the reactive properties (ignition and combustion velocity), the carbon-doped $Al/WO_3$ nanothermites exhibit slightly mitigated properties compared to the $Al/WO_3$ binary counterpart. Finally, this approach may be seen as providing the beginnings of an answer to the development of safer reactive nanothermites.

**Supplementary Materials:** The following are available online at http://www.mdpi.com/2311-5629/5/3/37/s1, Figure S1: FTIR spectra of the different $WO_3/C$ composites. The pristine $WO_3$ material is also reported. Figure S2: SEM view of the pristine $WO_3$ material.

**Author Contributions:** P.G. suggested and guided this research, and wrote the paper. Q.M., A.B. and P.G. carried out the experimental work including materials syntheses, the diverse characterisations (TGA, XRD, sensitivities and combustion tests) and the corresponding data interpretation. F.S. performed the electron microscopy coupled to the energy-dispersive X-ray spectroscopy. R.G. and D.S. supervised closely the study, the interpretation of data and helped in the proofreading of the manuscript.

**Funding:** The authors gratefully acknowledge the French Defence procurement agency (Direction Générale de l'Armement (DGA)), the Federal Office of Bundeswehr Equipment, Information Technology and In-Service Support (German Bundesamt für Ausrüstung, Informationstechnik und Nutzung der Bundeswehr (BAAINBw)) and the French National Centre for Scientific Research (CNRS) for their financial support.

**Acknowledgments:** The authors acknowledge S. Adach (SRSMC, University of Lorraine, France) and L. Vidal (IS2M, University of Mulhouse, France) for the elemental chemical and the transmission electron microscopy analyses, respectively.

**Conflicts of Interest:** The authors declare no conflicts of interest.

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
