# Peer review of "Mechanical Desensitization of an Al/WO3 Nanothermite by Means of Carbonaceous Coatings Derived from Carbohydrates"

_carbon, 2019_

Round 1

Reviewer 1 Report

The manuscript details the desensitization technique of Al-WO3 nanothermite using carbon addition through pyrolysis of sucrose and cellulose. The paper is interesting and relates to the very important topic of desensitization of nanothermites for safe handling and usage. There are minor changes and some questions that need to be addressed, namely:

In introduction (lines 37-38) you mention that nanothermites may substitute explosives in many pyrotechnic applications. This is true only if the nanothermites are powerful pressure generators (such as Al-I2O5 or Al-Bi2O3 nanothermites). However, addition of any desensitizers (carbon etc) drastically reduces the power of nanothermites (they still burn, but with much lower combustion wave speed and pressure generation abilities). Thus, first question: how would you expect the carbon desensitization you performed in this work, will affect the pressure generation abilities (you did not perform any confined burning experiments for pressure, only open, and no real combustion wave speed was measured). Al-WO3 gets desensitized, but maybe is becomes so "soft burning" and less powerful, that its practical application will be eliminated by this desensitization? Please address this concern in the text.

In line 341-342, you argue that the un-confinement of powder is required to release the gaseous products and thus maintain reactivity. This claim is not valid: the mixture Al-WO3 will burn in contained volume (will burn much better due to pressure increase), as it already contains all fuel and oxidizer it needs for complete burning (you used 1.4 equivalence ratio to take into account the oxide layer on Al). Thus, I think this sentence needs to be removed or changed appropriately.

There are several unclear sentences that needs to be revised, such as: 

a) line 20-21 (.. and have evidenced performing energetic mixtures)?

b) line 63 should be Maintaining

c) line 366 maintaining interesting (?) reactive properties (how interesting, maybe acceptable?), or line 373 (properties... are real and interesting(?), etc. Please scan the text for such unclear expressions.

Author Response

The manuscript details the desensitization technique of Al-WO3 nanothermite using carbon addition through pyrolysis of sucrose and cellulose. The paper is interesting and relates to the very important topic of desensitization of nanothermites for safe handling and usage. There are minor changes and some questions that need to be addressed, namely:

In introduction (lines 37-38) you mention that nanothermites may substitute explosives in many pyrotechnic applications. This is true only if the nanothermites are powerful pressure generators (such as Al-I2O5 or Al-Bi2O3 nanothermites). However, addition of any desensitizers (carbon etc) drastically reduces the power of nanothermites (they still burn, but with much lower combustion wave speed and pressure generation abilities). Thus, first question:

How would you expect the carbon desensitization you performed in this work, will affect the pressure generation abilities (you did not perform any confined burning experiments for pressure, only open, and no real combustion wave speed was measured).

In fact, initially, the combustion tests were performed in confined mode. For that, the different energetic compositions were poured in polymethylmethacrylate (PMMA) tubes and placed across from a flame igniter. The apparent densities of the energetic powders were approximately 10% ( loose powder) of their corresponding theoretical maximum densities (TMD). While a combustion velocity of 450 ± 30 m.sec-1 was determined for the binary Al/WO3 nanothermite, a drastic decrease of the combustion speed was recorded for both carbon-doped Al/WO3 energetic systems with values as low as ~ 2 m.sec-1. For us, two reasons can explain a decrease of the nanothermite reactivity:

. The addition of the non-energetic carbonaceous additive leads to the production of CO gas (2C(s) + O2(g) →  2CO(g)) during the exothermal reaction that may produce 25 – 30 cm3 STP of CO, taking into account the carbon amount in 0.5 g of the nanothermite composites. Because of the lower reactivity, the reaction front temperature is lower than that for a binary nanothermite. The gas flow at a moderate temperature and high velocity can increase the thermal transfer in the composite and decrease the hot point temperatures. This, in turn, leads to a decrease in the reaction rate.

. The amount of oxygen presents in the air and in the carbonaceous phase is not sufficient for the total combustion of the carbon. It is then possible that a reduction of WO3 into tungsten suboxides occurs at the carbon – oxide interphase. This could also lower the nanothermite reactivity.

For this same reason (low reactivity in tube experiments), no pressure measurements have been achieved. But, as suggested by the reviewer, the adding of carbon within nanothermites could decrease the pressure peak and then diverting their use to energetic applications where highly energy densities and pressures are not specifically required.

Al-WO3 gets desensitized, but maybe is becomes so "soft burning" and less powerful, that its practical application will be eliminated by this desensitization? Please address this concern in the text.

This remark is justified and we have added such a comment (Introduction section, lines 78-84) in the new version of the manuscript.

“Generally speaking, the adding of additives within nanothermites leads to a mitigated reactive behaviour of the considered energetic systems (decrease of the combustion velocity, longer ignition delay time…) that can thereby limit or divert the thermites of their initial practical applications but in other side, it is important to keep in mind that using nanothermites free of any additives can be extremely hazardous (handling and transportation). Definitively, the nanothermite desensitization is an emerging research topic where numerous aspects must be explored (additive nature, combustion behaviour and potential applications).”

In line 341-342, you argue that the un-confinement of powder is required to release the gaseous products and thus maintain reactivity. This claim is not valid: the mixture Al-WO3 will burn in contained volume (will burn much better due to pressure increase), as it already contains all fuel and oxidizer it needs for complete burning (you used 1.4 equivalence ratio to take into account the oxide layer on Al).

Thus, I think this sentence needs to be removed or changed appropriately.

In a first time, we wanted to add some informations about the combustion tests performed in confined mode (PMMA tube experiments as discussed in the previous remarks) but for the sake of clarity and to avoid any confusion, we have decided to delete this sentence.

There are several unclear sentences that needs to be revised, such as:

a) Line 20-21 (…and have evidenced performing energetic mixtures)?

The sentence “The reactive properties of the carbon-based energetic mixtures were also evaluated and have evidenced performing energetic mixtures.” was modified as follows: “The combustion behaviour of the carbon-based energetic mixtures were also evaluated and a respectable reactivity has been evidenced”

b) Line 63 should be “Maintaining”

The corresponding sentence was rewritten as requested by the reviewer (line 64).

c) Line 366 maintaining interesting (?) reactive properties (how interesting, maybe acceptable?), or line 373 (properties…are real and interesting (?), etc. Please scan the text for such unclear expressions.

These remarks have been taken into account into the revised version of the manuscript.

Line 366 à line 377: the word “interesting” was replace with “acceptable”.

Line 373 à line 383: the sentence “The reactive properties (ignition and combustion velocity) of the carbon-doped Al/WO3 nanothermites are real and interesting, even if a slight mitigation seems to be observed…” was rewritten as follows: “With regard to the reactive properties (ignition and combustion velocity), the carbon-doped Al/WO3 nanothermites exhibit slightly mitigated properties compared to the Al/WO3 binary counterpart.”

Reviewer 2 Report

In this very good paper the Authors presented novel method for desensitization of nanothermites. It is a great pleasure to read such well prepared manuscript. Extensive use of technques like SEM, TEM, X-Ray diffraction, and thermogravimetry provided a lot of insight into the physicochemical properties of the composites. Moreover, the mechanical tests, namely impact and friction sensitivity confirmed the usefulness of both sucrose and cellulose coatings, with only a minimal impact on burning speed. What is important, also the preparation procedures were described very accurately, which can easily facilitate their repetition by other researchers.

I have only one question: where is equation 5, mentioned in line 346? I suggest to put in in the text of the manuscript.

Author Response

In this very good paper the Authors presented novel method for desensitization of nanothermites. It is a great pleasure to read such well-prepared manuscript. Extensive use of techniques like SEM, TEM, X-Ray diffraction, and thermogravimetric provided a lot of insight into the physicochemical properties of the composites. Moreover, the mechanical tests, namely impact and friction sensitivity confirmed the usefulness of both sucrose and cellulose coatings, with only a minimal impact on burning speed. What is important, also the preparation procedures were described very accurately, which can easily facilitate their repetition by other researchers.

I have only one question: where is equation 5, mentioned in line 346? I suggest to put in in the text of the manuscript.

This remark has been taken into account in the revised version of the manuscript. The corresponding equation (5) has been added: 2C(s) + O2(g)   2CO(g).
